# Spatial Characteristics Analysis for Coupling Strength among Air Pollutants during a Severe Haze Period in Zhengzhou, China

**DOI:** 10.3390/ijerph19148224

**Published:** 2022-07-06

**Authors:** Linan Sun, Antao Wang, Jiayao Wang

**Affiliations:** 1College of Resources and Environment, Henan University of Economics and Law, Zhengzhou 450046, China; 2College of Geography and Environmental Science, Henan University, Kaifeng 475004, China; jfzx@hnp.edu.cn; 3Department of Cyber Security, Henan Police College, Zhengzhou 450046, China; wat@hnp.edu.cn

**Keywords:** spatial interpolation, CDFA, multifractality, coupling correlation, coupling contribution

## Abstract

This paper investigates the multifractal characteristics of six air pollutants using the coupling detrended fluctuation analysis method. The results show that coupling correlations exist among the air pollutants and have multifractal characteristics. The sources of multifractality are identified using the chi square test. The coupling strengths between different pollutants are quantified. In addition, the coupling contribution of a series in the haze system is calculated, and SO_2_, as the main pollutant, plays a key role in the pollution system. Moreover, the Kriging interpolation method is used to analyze the spatial characteristic on coupling contribution of SO_2_. The spatial analysis of coupling strength for air pollutants will provide an effective approach for pollution control.

## 1. Introduction

With the rapid development of China’s economy, industrialization and urbanization are progressing rapidly, and environmental problems have become increasingly serious in China since the 21st century. Especially, the persistent haze appears in a large scale and high frequency [1,2] in recent years, which has become an important environmental problem. The haze pollution has aroused widespread attention in the scientific community [3].

The pollutants time series are nonstationary due to the trends that exist in the haze systems. In order to find the correct scaling behavior of intrinsic fluctuations in these systems, one must separate them from the trends. In 1994, Peng et al. (1994) [4] introduced detrended fluctuation analysis (DFA) that detects the long range correlations in the nonstationary time series. In 2002, Kantelhardt et al. (2002) [5] introduced multifractal detrended fluctuation analysis (MF-DFA), an extended version of DFA, to analyze the multifractal properties of time series. These methods have been widely applied to study the statistical properties of a single time series, such as the time series in finance stocks (Ausloos, 2000) [6], heart rate dynamics [7] and precipitation [8]. Many scholars have used different methods to study the air pollutants time series, such as Lee et al. (2002, 2003) [9,10], Shi et al. (2008, 2011, 2015) [11,12,13], Muñoz et al. (2013) [14], Shi et al. (2008, 2011, 2015) [11,12,13], Muñoz et al. (2013) [14], Shen et al. (2016) [15], Zhu et al. (2010) [16] and Tong et al. (2007) [17].

Some studies explored the cross-correlations between non-stationary financial time series using the detrended cross-correlation method (DCCA) proposed by Podobnik et al. (2008) [18]. Subsequently, Zhou et al. (2008) [19] combined the multifractal detrended fluctuation analysis and DCCA method (the combination of which is known as the MF-DCCA method) to investigate multifractal cross-correlations. The MF-DCCA method, based on q-order detrended covariance [20,21,22,23], is a combination of multifractal analysis and detrended cross-correlation analysis. The DCCA and MF-DCCA methods have been widely applied in many fields, such as financial data [24,25], traffic flow [26,27], river fluctuation [23] and meteorological data [28]. Many scholars have used different methods to study the air pollutants time series [29,30,31,32,33,34,35,36].

The DFA and MF-DFA methods are used to study the statistical properties of a single nonstationary time series, and the DCCA and MF-DCCA methods are used to study the multifractal behavior between two time series of one dimension or higher. However, in the real world, there are many cases in which more than two time series are correlated to each other. Therefore, these methods cannot provide us more complete information about more than two time series, and they cannot consider the model capabilities of more parameters at the same time. In order to investigate the coupled correlations among more than two stationary time series, Hedayatifar et al. (2011) [37] proposed the coupling detrended fluctuation analysis (CDFA) method (an extended version of DCCA) that analyze the coupling correlation over two sets of series. This method can identify the sources of multifractality.

The CDFA method has been used in the study of time series in various fields in recent years. Sun (2017) [38] and Wang et al. (2019, 2022) [39,40], respectively, used the CDFA method to study the multifractal characteristics of air pollution in several important cities in China such as Beijing and Zhengzhou and obtain many meaningful conclusions. Yao et al. (2017, 2020) [41,42] studied the coupling elimination fluctuation trend of warehouse quantity and the nonlinear correlations among economic policy uncertainty (EPU), the crude oil market and the stock market by using the CDFA method. So far, the application of the CDFA method is not too much, especially in the research of air pollution, which can be studied more widely and deeper.

From 24:00 17 December 2016 to 23:00 26 December 2016, Zhengzhou suffered the most serious haze pollution in its history and was the most polluted area in the country at that time. This paper takes the typical haze pollution event as the research object. The methods of CDFA and spatial analysis are used to explore the main pollution sources and analyze the causes of air pollution, which can provide some scientific basis for the prevention and control of air pollution.

This paper will take advantage of the CDFA approach to study six kinds of typical pollutants in Zhengzhou severe haze weather and analyze coupling strength between pollutants, as well as their contribution to the haze power system. In particular, the spatial interpolation is combined with the CDFA method in a uniquely interdisciplinary approach to produce more meaningful spatial knowledge and information by spatial distribution characteristics of coupling contribution of pollutants.

Compared with the existing literature, our main contributions are as follows:In this paper, the coupling relationship of six pollutants in a severe haze in Zhengzhou City from 17 December 2016 to 26 December 2016 was studied by using CDFA method. The coupling strength between different pollutants are quantified in the haze system, and the contribution of one of the series in the coupling of the others, are quantified and analyzed.The coupling contribution of the main pollutants in the haze system is studied via spatial interpolation, and the spatial characteristics of the pollutants are analyzed according to the spatial distribution. The spatial analysis of coupling intensity of air pollutants provides an effective method for exploring the causes, so as to provide some scientific basis for pollution control.

This paper is organized as follows: Section 2 introduces the research area and data. Section 3 outlines the methodology of CDFA. Section 4 analyzes the air pollutants using the method of DCCA, and the empirical results are obtained. Section 5 analyzes the spatial characteristics for coupling contribution of SO_2_ using spatial interpolation method. Section 6 presents the conclusions.

## 2. Research Area and Data Description

Zhengzhou City is located in 112°42′~114°14′ E, 34°16′~34°58′ N, the north-central part of Henan Province. To the north of Zhengzhou is the Yellow River, to the west is Songshan, and to the southeast is the Huang Huai plain, as seen in Figure 1.

As the provincial capital of Henan Province, Zhengzhou is an important city on the new Eurasian Continental Bridge, a national central city and a historical and cultural city. The atmospheric environment in Zhengzhou presents a significant compound pollution situation in recent years. Under the combined effects of industrial pollution, motor vehicle emission, building activities and life area source, etc., fine particle pollution in the atmosphere in Zhengzhou is increasing. In 2016, severe pollution weathers frequently happened in Zhengzhou, the scope of regional pollution has expanded, and the characteristics of compound pollution in large cities have been becoming increasingly prominent. What’s more, air pollution is more severe during the winter heating season, from late 2016 to early 2017, the severe regional haze pollution for many days caused by high concentrations of pollutants happened again and again. The haze pollution can cause pulmonary sclerosis, asthma and bronchitis, even cardiovascular disease. The general public have attached great importance to haze pollution. How to propose effective control measures on haze weather is a difficult problem to be solved at present. More profound study is needed on the prevention and control of air pollution in Zhengzhou.

There are nine automatic monitoring stations (as seen in Figure 1), including the Gang Li Reservoir (GLR) located in Huiji district, Water Supply Company (WSC) located in Gaoxin district, First Affiliated Hospital of Zhengzhou University (FAHZU) located in Erqi district, Administrative Committee of Economic Development Zone (ACEDZ) located in Jingkai district, Monitoring Station of Zhengzhou (MSZZ) located in Zhongyuan district, The 47th Middle School (MS47) located in Zhengdong New district, Tobacco Factory (TF) located in Guancheng district, Banking Institute (BI) and Zhengzhou Textile Machinery Limited Company (ZTMLC) located in Jinshui district.

Near the MSZZ and TF stations are Zhengzhou railway station, Zhengzhou residential and traffic intensive areas, where there is a thermal power plant next to the MSZZ station. The TF, BI and MS47 stations are mainly surrounded by schools, parks and residential areas, and they are close to the heat source plant. Near the ZTMLC station, there are some enterprises with serious pollution, such as a porcelain factory, Hengtai aluminum alloy factory, Zhengzhou first steel plant and Yuxiang electric power engineering company, etc. Around the FAHZU and BI stations, there are mainly residential areas, hospitals and schools, and they are far away from the industrial pollution sources. The GLR station is mainly surrounded by reservoirs and Yellow River scenic spots. The WSC station is close to the industrial zone and heat source plant in the western of Zhengzhou. Around the ACEDZ station, there are mainly residential areas and some industrial areas, and it is close to the famous Zhongyuan futa. These urban monitoring stations are evenly distributed in the important areas of Zhengzhou, and the pollution data from that can well reveal the whole pollution situation about Zhengzhou.

The hourly data of NO_2_, CO, O_3_, PM_10_, PM_2.5_ and SO_2_ in Zhengzhou at nine automatic monitoring stations from 24:00 17 December 2016 to 23:00 26 December 2016, are supplied by the China National Environmental Monitoring Centre [43] in this paper. The missing data have been replaced by the mean between two of the nearest data. In order to learn the general situation during the severe haze period in Zhengzhou, the average value of each air pollutant time series at nine stations is calculated, as shown in Figure 2. As seen from Figure 2, the haze continues for about 9 days (about 240 h), and it is the severest haze pollution in Zhengzhou in 2016, which the average maximum of PM_2.5_ is close to 800 μg/m^3^.

## 3. Introduction to CDFA Method and Multifractal Contribution Sources

### 3.1. CDFA (Coupling Detrended Fluctuation Analysis) Method

To reveal the multi fractal characteristics of CO, NO_2_, O_3_, PM_10_, PM_2.5_ and SO_2_ in Zhengzhou, we use the CDFA approach proposed by Hedayatifar et al. [37]. The method is well suited for testing the mutual coupling of over two series, and it can be described as follows:

Series containing *n* time series {$x_{m}^{1}$,…, $x_{m}^{j}$,…, $x_{m}^{n}$} in *N* is considered, and, *m* is the member of each time series. Constructing cumulative series as:(1)$$X^{j}(i)=\sum_{m=1}^{i}(x_{m}^{j}-<x^{j}>)$$
where *i* = 1, 2, *N*, *j* = 1, 2, *n*, $<x^{j}>=\frac{1}{N}\sum_{m=1}^{N}x_{m}^{j}$.Cumulative series $X^{j}(i)$ is divided into *N* boxes not overlapping each other in *s* length, and $N_{s}=\mathrm{int}(N/s)$, as *N* is often not be an integral multiple of *s*, cumulative column tail may have partial residual data not for calculation. in order to take into account such remaining data, the partitioning process can be repeated at the tail of the cumulative series, therefore, a total of 2*N_s_* small boxes have been obtained.Then, the least square method is used to fit the cumulative series in each small box to gain the local trend $x_{\lambda }^{j}(i)$ of 2*Ns* boxes. Then the detrended multicovariance is calculated as
(2)$$\left\{ \begin{array}{l} F^{\lambda }(s)=\frac{1}{s}\sum_{i=1}^{s}\prod_{j=1}^{n}\left|X^{j}\left[(\lambda -1)s+i\right]-x_{\lambda }^{j}(i)\right|\begin{array}{cc} , & \end{array}\lambda =1,2,\ldots ,N_{s}\\ F^{\lambda }(s)=\frac{1}{s}\sum_{i=1}^{s}\prod_{j=1}^{n}\left|X^{j}\left[N-(\lambda -N_{s})s+i\right]-x_{\lambda }^{j}(i)\right|\begin{array}{cc} , & \end{array}\lambda =N_{s}+1,2,\ldots ,2N_{s} \end{array}\right.$$The *q*th order fluctuation function is calculated as follows:(3){Fxm1,…,xmn(q,s)≡{12Ns∑λ=12Ns|Fλ(s)|qn}1q,q≠0Fxm1,…,xmn(q,s)=exp{12nNs∑λ=12Nsln|Fλ(s)|},q=0If time series $x_{m}^{1}$,…, $x_{m}^{j}$,…, $x_{m}^{n}$ is involved in long-range power law correlation, the $F_{x_{m}^{1},\ldots ,x_{m}^{n}}(q,s)$ will increase as a power law of *s*.
(4)$$F_{x_{m}^{1},\ldots ,x_{m}^{n}}(q,s)∼s^{h_{x_{m}^{1},\ldots ,x_{m}^{n}}(q)}$$
where $h_{x_{m}^{1},\ldots ,x_{m}^{n}}(q)$ is the generalized CDFA scaling exponent for all the time series, which can be obtained by calculating the slope of the log-log plots of $F_{x_{m}^{1},\ldots ,x_{m}^{n}}(q,s)$ versus *s*.

When *n* = 1, *q* = 2, it is simplified to the DFA method, which can be used to study the long-term persistence characteristics of only one time series. When *n* = 2, *q* = 2, it is known as the DCCA method, and the method can be used to analyze the correlation between two time series. If $h_{x_{m}^{1},\ldots ,x_{m}^{n}}(q)$ *= c*, *c* is a constant*,* then the coupling relation is monofractal, while its value changing along with *q* indicates the presence of multifractality. More specifically, if $h_{x_{m}^{1},\ldots ,x_{m}^{n}}(q)$ > 0.5, the coupled correlation is positive, and it means that a large (or small) increment in one variable is more likely to be followed by a large (or small) increment in other variables. If $h_{x_{m}^{1},\ldots ,x_{m}^{n}}(q)$ < 0.5, the coupling correlation is negative, i.e., anti-correlated, which means that a large (or small) increment in one variable is more likely to be followed by a large (or small) decrement in other variables. If $h_{x_{m}^{1},\ldots ,x_{m}^{n}}(q)$ = 0.5, there is no coupling correlation among time series, and the changes in one variable cannot affect the behavior of other variables.

In addition, for *q* > 0, *h*(*q*) describes the scaling behaviour of the segments with large fluctuations. For *q* < 0, *h*(*q*) describes the scaling behaviour of segments with small fluctuations. Therefor the multifractality gives information about the relative importance of various fractal exponents present in the time series. The strength of the multifractality of a time series can be characterized by Δ*h* = *h*_max_(*q*) − *h*_min_(*q*), where Δ*h* is the range of generalized Hurst exponents *h*(*q*). The larger the Δ*h* is, the stronger the multifractality degree become, and vice versa.

### 3.2. Multifractal Contribution Sources

Researchers have shown interest in the nature of multifractal behavior in time series. There are two multifractal contribution sources which can be found in the time series [44,45]. One is the cross-correlation of different fluctuations, and the other is the variable fat-tailed probability distribution [46]. The two sources can be realized using a shuffling procedure (shuffled data) and phase randomization procedure (surrogate data), so as to study the multifractal characteristics of time series. The shuffling procedure can eliminate cross-correlation of original data, and keep the original data distribution unchanged. Thus, the shuffled data can be used to study multifractal contribution from cross-correlation in time series. On the contrary, the phase randomization procedure can eliminate non-Gaussian distribution and only retain correlation of original data. Thus, the surrogate data can be used to study the multifractal contribution of variable fat-tailed probability distribution in time series.

According to the chi square statistical test method proposed by Hedayatifar et al. [37], We can quantify the coupling strength of long-term memory or fat-tail probability as follows:(5)$$\chi_{⋄}^{2}(Y)=\sum_{i=1}^{N}\frac{{\left[h(q_{i})-h_{⋄}^{Y}(q_{i})\right]}^{2}}{\sigma {\left(q_{i}\right)}^{2}+\sigma_{⋄}^{Y}{\left(q_{i}\right)}^{2}}$$
(6)$$\left\{ \begin{array}{l} \sigma (q)={\left\{\frac{1}{N}\sum_{i=1}^{N}\prod_{j=1}^{n}{\left|X^{j}(i)-\langle X^{j}\rangle \right|}^{\frac{q}{n}}\right\}}^{\frac{1}{q}}\begin{array}{cc} & \;\;\;q\neq 0 \end{array}\\ \sigma (q)=exp\left\{\frac{1}{nN}\sum_{i=1}^{N}ln\left(\prod_{j=1}^{n}\left|X^{j}(i)-\langle X^{j}\rangle \right|\right)\right\}\begin{array}{cc} & q=0 \end{array} \end{array}\right.$$

The “⋄” symbol represents “surrogate” or “shuffled”, and the *Y* means that the series has been phase randomized or shuffled. We can use the following local Relation (7) to quantify the contribution of sequence y in the other coupling.
(7)$${\bar{\chi }}_{⋄}^{2}(Y)=100\cdot \chi_{⋄}^{2}(Y)/\chi_{⋄}^{2}(all)$$
where
(8)$$\chi_{⋄}^{2}\left(all\right)=\sum_{i=1}^{N}\frac{{\left[h(q_{i})-h_{⋄}^{all}(q_{i})\right]}^{2}}{\sigma {\left(q_{i}\right)}^{2}+\sigma_{⋄}^{all}{\left(q_{i}\right)}^{2}}$$

The “*all*” specifies all the series are shuffled, and the σ(q) is the generalized standard deviation for *q*.

## 4. Coupling Detrended Fluctuation Analysis (CDFA) of Air Pollutants Series

In order to study the multifractal characteristics and quantify the coupling correlations of CO, NO_2_, O_3_, PM_10_, PM_2.5_ and SO_2_, we first need to process the original data of six pollutants by shuffling and surrogate procedures, respectively. Figure 3 shows the result surfaces obtained from Formula (4) for original time series (Figure 3a), shuffled time series (Figure 3b) and surrogate time series (Figure 3c) of the six pollutants. It should be stated that the data used in Figure 3 are the average data of six pollutants (i.e., the data expressed in Figure 2). It can be seen from Figure 3 that there exist power-law coupled correlations from the log-log plots of fluctuation functions $F_{x_{m}^{1},\ldots ,x_{m}^{n}}(q,s)$ versus *s* for all three figures.

It can be seen that the surface of Figure 3b is more wrinkled and less smooth than that of Figure 3a. And as seen from Figure 3b, the value of $h_{x_{m}^{1},\ldots ,x_{m}^{n}}(q)$, which can be obtained by calculating the slope of $ln(F_{x_{m}^{1},\ldots {,}_{m}^{n}}(q,s))$ versus ln(s) plots, is near 0.5, and it proves the conclusion mentioned in Section 3 that the shuffling procedure eliminates cross-correlation of original data and keeps original data distribution unchanged. In addition, the surface in Figure 3c becomes more smoother than that in Figure 3a,b, and it means that the surrogate procedure eliminate non-Gaussian distribution and retain correlation of original data.

Based on the above conclusions and analysis, the shuffled and surrogate processing for the six pollutants original data in this section are very effective. Additionally, the shuffled and surrogate data can meet the requirements of CDFA multifractal analysis in this paper well.

### 4.1. Analysis on Multifractal Contribution Sources of Pollution Series

According to Section 3.2, there are two sources, i.e., cross-correlation and variable fat-tailed probability distribution, which can affect the multifractal characteristics of time series. However, for a specific system, the two sources may not all have an important influence on the multifractal. So, we first analyze which source has a greater impact on the haze pollution system studied in this paper, and we also prepare for the analysis on coupling correlation and contribution of six pollutants in next section.

The steps of calculation and analysis in this section are: First, the time series of CO, NO_2_, O_3_, PM_10_, PM_2.5_ and SO_2_ at nine stations are processed by shuffling and phase randomization procedures, respectively, then we can obtain the shuffled and surrogate series of them. Second, Using CDFA method to calculate $h_{x_{m}^{1},\ldots ,x_{m}^{n}}(q)$ (simplify for *h*(*q*)) values versus *q* of the six pollutants series based on the Formula (4) in Section 3.1, and plot the *q~h*(*q*) curves as shown in Figure 4. Finally, by comparing the changes of shuffled and surrogate series curves relative to the original series curve, it can be concluded that the curve with greater changes is the source with greater impact on multifractal.

In order to more accurately study the influence of different sources on multifractal, the behaviors of *q~h*(*q*) of original series, shuffled series and surrogate series of six pollutants at nine stations are given using CDFA method in Figure 4, and the *q~h*(*q*) curves are calculated by the Formula (4). The black line with “★” is the CDFA analysis on the original series of six pollutants. The red line with “◆” is the CDFA analysis on the shuffled series of six pollutants, and the blue line with “●” is the CDFA analysis on the surrogate series of six pollutants.

As shown in Figure 4, the CDFA analysis of the shuffled series is significantly different from that of the original series at nine stations. It shows that the shuffling procedure is effective in cleaning data correlation, and the *h*(*q*) value of shuffled series is close to 0.5. This implies that almost all the series after being shuffled have no coupling correlation, and shows again that it is very effective to study the coupling strength and contribution of each pollutant by shuffling procedure.

As shown in Figure 4, the behaviors of *q~h*(*q*) for shuffled and surrogate series are different from the original series. All the *q~h*(*q*) curves of surrogate series are close to the original series; however, the *q~h*(*q*) curves of shuffled series are far away from the original series. It is obvious to see from Figure 4 that there is $h_{shuff}(q)<h_{surr}(q)$ relationship for each monitoring station. This means that during the period of severe haze, the mutual coupling of pollutants time series is not only affected by cross-correlation, but also affected by the leptokurtic and fat tailed feature. The cross-correlation play a dominant role in the coupling relationship among the pollutants time series. The coupling correlation between pollutants is mainly caused by the large and small fluctuations, while the leptokurtic and fat tailed effect for the peak value are not large. Therefore, the influence of cross-correlation for coupling strength of pollutants is analyzed in the next section.

### 4.2. Coupling Correlation and Contribution of Pollution Series in the Haze System

From the analysis in the above section, we have learned that the multifractal source of this severe haze pollution in Zhengzhou is cross-correlation. Let’s take SO_2_ as an example to explain the analysis process of the CDFA method. If we want to study the coupling contribution of SO_2_ in the haze system using the CDFA method, we can take the following steps:(1)Processing original time series of SO_2_ (named as Origin_SO_2_) by shuffling procedure, and we obtain the shuffled time series of SO_2_ (named as Shuff_SO_2_).(2)Substituting the Shuff_SO_2_, Origin_CO, Origin_NO_2_, Origin_O_3_, Origin_PM_10_, Origin_PM_2.5_ into Formula (4) of Section 3.1, and we calculate the value of *h(q)* versus *q.*(3)Plotting the *q~h*(*q*) curve, we name the curve as CDFA_SO_2_, which represents the generalized scaling exponents of CDFA when only SO_2_ is shuffled. (CDFA_CO, CDFA_NO_2_, CDFA_O_3_, CDFA_PM_10_, and CDFA_ PM_2.5_ can be calculated in the same way).(4)Replace the Shuff_SO_2_ with Origin_SO_2_ in step (2) above, and we calculate the *q~h*(*q*) again. Plotting the new *q~h*(*q*) curve which named CDFA_Origin, it denotes the generalized scaling exponents of CDFA for all the original time series.(5)The coupling contribution of SO_2_ (or CO, NO_2_, O_3_, PM_10_, PM_2.5_) can be analyzed by using the relevant parameters in Section 3 finally.

We calculate and plot the *q~h*(*q*) curves of CDFA_Origin, CDFA_CO, CDFA_NO_2_, CDFA_O_3_, CDFA_PM_10_, and CDFA_PM_2.5_ at nine stations, respectively, according to the above steps as seen in Figure 5. It shows the behaviors of *h*(*q*) when applying the CDFA for CO, NO_2_, O_3_, PM_10_, PM_2.5_ and SO_2_ time series, only the mentioned time series is shuffled and the other original time series are kept unchanged. By using the shuffling procedure, only the correlation effect is washed out and the probability density function effect exists.

From Figure 5, we can see that the *h(q)* value changing along with *q* indicates the presence of multifractality. Meanwhile, the shuffled series shows distinct differences for q<0 and q>0, that is, the coupling strength among the pollution series shows distinct differences for the small fluctuations (q<0) and the large fluctuations (q>0) in the original series. For example, in the GLR and ACEDZ stations, when q<0, the O_3_ is the farthest pollutant from the original series after being shuffled, which illustrates that it has the largest influence in small fluctuations of the haze system; when q>0, it is the nearest pollutant from the original series after being shuffled. This illustrates that O_3_ has the least influence in large fluctuations of the haze system. For SO_2_, the influence of SO_2_ in q>0 is obvious greater than that in q<0, and this indicates that the SO_2_ has a greater influence in the large fluctuations, and has a less influence in the small fluctuations of the haze system. The similar conclusions can be obtained for other pollutants as well.

We can quantify the coupling strength and contribution of cross-correlation by calculating the values of $\chi_{shuff}^{2}$ and ${\bar{\chi }}_{shuff}^{2}$ make use of Formulas (5) and (7). Table 1 gives the average of $\chi_{shuff}^{2}$ and ${\bar{\chi }}_{shuff}^{2}$ values for six pollutants in nine stations. To be convenient for analysis, the values are expressed by the high-dimension graph for 0≤q≤20, as shown in Figure 6.

As seen from Figure 6a,b, in the ACEDZ, MSZZ and MS47 stations, the $\chi_{shuff}^{2}$ and ${\bar{\chi }}_{shuff}^{2}$ values of PM_10_ are the largest, which indicates that the coupling correlation and contribution of PM_10_ with other pollutants in large fluctuation is the greatest. In the GLR and BI stations, the pollutants with the largest $\chi_{shuff}^{2}$ and ${\bar{\chi }}_{shuff}^{2}$ values are CO and SO_2_ respectively, that is, the CO and SO_2_ have the strongest coupling correlation and largest coupling contribution, as observed from Table 1. In terms of CO, NO_2_, O_3_, PM_10_, PM_2.5_ and SO_2_ at the nine stations, the average values of $\chi_{shuff}^{2}$ and ${\bar{\chi }}_{shuff}^{2}$ are 5.678 × 10^−4^, 5.256 × 10^−4^, 8.453 × 10^−5^, 8.980 × 10^−4^, 8.182 × 10^−4^, 6.221 × 10^−4^ and 1.88, 1.751, 0.29, 3.008, 2.719, 2.115, respectively. It means that the coupling correlation of PM_10_, PM_2.5_ and SO_2_ are stronger, and the coupling contribution of them are greater than that of other pollutants in the haze system.

Based on the above analysis, SO_2_ has the greatest impact in the haze system other than PM_10_ and PM_2.5_. SO_2_ not only pollutes the atmosphere in gaseous form, but more than 30% of it can be converted into sulfate aerosols to cause an increase of atmospheric particulate concentration. It is also a critical precursor reactant for secondary aerosol. In the winter, the increase of coal emissions results the increase of SO_2_ emissions, while the SO_2_ is the key gaseous precursor of secondary inorganic aerosol, so it directly affects the large fluctuation part in the haze system, such as the violent fluctuations of PM_2.5_. Therefore, the SO_2_ has the largest impact and plays a dominant role in the large fluctuation part of the haze coupling system except PM_10_ and PM_2.5_. The finding is consistent with the source apportionment experiment of PM_2.5_ in Zhengzhou [47,48,49]. In 2010, the dominant components of PM_2.5_ in Zhengzhou are secondary ions (sulfate and nitrate) and carbon fractions, which soluble ions and total carbon contribute 41% and 13% of PM_2.5_ mass, respectively [47]. Jiang et al. (2017) [48] conduct the sample source apportionment experiment of PM_2.5_ during the periods from 2012 to 2015 in Zhengzhou, and Chemical Mass Balance results show that the contributions of major sources (i.e., nitrate, sulfate, biomass, carbon and refractory material, coal combustion, soil dust, vehicle, and industry) of PM_2.5_ are 13%, 16%, 12%, 2%, 14%, 8%, 7%, and 8% in heavily polluted days, with sulfate and nitrate as major ions. In 2016, the sum of the concentrations of ${\mathrm{SO}}_{4}^{2-}$, NO^3−^, and NH^4+^ increase with the aggravation of pollution level, and the secondary aerosols is the main source of PM_2.5_, which accounts for 38.4% on heavy pollution days in Zhengzhou [49].

## 5. Spatial Characteristics Analysis on Coupling Strength of Air Pollutants

Through the above analysis on coupling parameter values in each station, it can be found that they have obvious spatial characteristics. Therefore, it is necessary for us to make spatial analysis on pollutant coupling strength parameters to mine more useful conclusions. The coupling correlations and contributions of SO_2_ is the greatest for this haze among the four gaseous pollutants except PM_10_ and PM_2.5_. The PM_10_ and PM_2.5_ are fine particles, and their formation are diverse and complex. However, SO_2_ mainly comes from the direct emission of pollution sources, and its emission is also a cause of the formation for PM_10_ and PM_2.5_. SO we think it is more meaningful to study the spatial characteristics of SO_2_ coupling strength.

In the severe haze of Zhengzhou, the cross-correlation has stronger impact than the fat tailed distribution on multifractal characteristics for pollutants. In addition, the coupling contribution is good enough to characterize the coupling strength of pollutants in the haze power system, so we use ArcGIS software to analyze the spatial distribution of the average concentration and coupling contribution ${\bar{\chi }}_{shuff}^{2}$ value of SO_2_ in the nine monitoring stations by using Kriging interpolation. The spatial distribution of the average concentration and ${\bar{\chi }}_{shuff}^{2}$ value of SO_2_ in Zhengzhou city by contour interpolation method when 0≤q≤20 is given in the Figure 7a,b.

As seen in Figure 7a, the spatial distribution of SO_2_ concentration in Zhengzhou presents decreasing trends from the central zone to north and south ends. Firstly, the west and east of the city are factory concentrated areas, and factory exhaust emissions are a major source of SO_2_ pollutant; secondly, the area is also a densely populated area, and the winter is a heating period. Therefore, the pollution gases emitted by traffic, coal burning -are also an important source of SO_2_; Finally, from Figure 7a, it can be seen that the production of SO_2_ should be closely related to the distribution of thermal power plants, heat source plants and waste disposal plants. Throughout the western region, especially in the ZTMLC, there exists maximum SO_2_ concentration. On the one hand, the western region is an industrial concentration area, on the other hand, the heavily polluting enterprises, such as City porcelain plant, City Hengtai aluminum alloy factory, Zhengzhou No.1 steel mill, Yuxiang power engineering company are distributed near the ZTMLC, and this is also the direct cause of the serious SO_2_ pollution. Although the actual concentration of SO_2_ in the eastern and northern parts of the city is not small as well, this may be related to winter heating by coal combustion and motor vehicle exhaust emissions, and the emission sources are relatively less than those in the western region, so the SO_2_ concentration is relatively low here.

As seen from Figure 7b, the spatial distribution of SO_2_ coupling contribution in Zhengzhou presents a decreasing trend from central and southeast regions (FAHZU, BI and ZTMLC) to northwest region. The spatial distribution is different from that of SO_2_ concentration, and this indicates that the concentration of SO_2_ is high, and its coupling contribution may not be necessarily large, which may also be affected by the weather and chemical reaction between pollutants. That is, there may be many factors influence the coupling chemical reaction of SO_2_. As seen from the Figure 7b, in the vicinity of ZTMLC, the concentration of SO_2_ is very large where coupled chemical reactions are more likely to happen, thus resulting in the relatively large coupling contribution in this area. In the vicinity of FAHZU and BI, the actual concentration value is relatively small, because it is the main residential areas, hospitals and schools, away from industrial pollution sources. However, nearby the FAHZU, as seen from Figure 6b, the coupling contribution of NO_2_ is the least, the regional haze is mainly caused by SO_2_, and the SO_2_ plays a main role in the haze system. As seen also from Figure 6b, the maximum value of coupling contributions for NO_2_, SO_2_, PM_2.5_ and PM_10_ appear in the BI station, which indicates that in this area these pollutants are involved in a series of coupling chemical reactions. The pollutants of NO_2_, SO_2_ and O_2_ produce nitric acid, sulfuric acid and O_3_ under the light and chemical actions, so the coupling contribution of O_3_ is maximum. What’s more, the nitric acid and sulfuric acid produced by SO_2_ form aerosols, then convert to PM_2.5_ and PM_10_, therefore, the coupling contribution of both PM_2.5_ and PM_10_ have a local maximum. During this haze, the ground wind in winter mainly tends north and northwest in Zhengzhou, and it has a positive effect on the coupling of SO_2_ with other pollutants in the Southeast. It makes the SO_2_ have a great coupling contribution in the south central Zhengzhou. In the GLR station, the coupling contribution of SO_2_ is minimum. This is mainly because the area around the GLR station are reservoirs, as well as the Yellow River scenic spots, and there are no sources of SO_2_ emissions. As seen from Figure 7a, the SO_2_ concentration is also the lowest due to the small coupling contribution of SO_2_ in the station.

## 6. Conclusions

In this paper, the coupling relationship among six pollutants (CO, NO_2_, O_3_, PM_10_, PM_2.5_ and SO_2_) in a severe haze weather from 17 December 2016 to 26 December 2016 in Zhengzhou is analyzed by the CDFA method, as well as the effects of various factors on city haze pollution. The CDFA method can be used to quantify the strength of the coupling correlations and their coupling contributions of different pollutants in haze weather, thus a detailed understanding on the role and status of different pollutants in the haze power system can be gained.

From the experimental results, we can see that the SO_2_ has the greatest influence in the large fluctuation part of haze coupling system. It is concluded that the primary pollutant of SO_2_ plays a very important role in the typical haze in Zhengzhou, and it is the main contribution to the haze system. This is consistent with that Zhengzhou City belongs to SO_2_ Pollution Control Zone. Thus, the coupling contribution of SO_2_ in the haze system is studied by spatial distribution using Kriging interpolation, and the spatial characteristics of the pollutants are analyzed according to the spatial distribution.

The spatial analysis on coupling intensity of air pollutants will provide a good approach for analyzing pollution sources. It is noteworthy that there is an increasing trend for the NO_2_ pollution in 2016, and the contribution of NO_2_ pollutants to the haze cannot be ignored. The spatial interpolation on coupling contribution of pollutants in each station is studied. It is a new attempt and innovation to study the distribution of coupling contribution of pollutants in each station by spatial interpolation method. Through spatial analysis, we can obtain the spatial distribution of pollutants concentration or relevant parameter values, and further analyze the causes of air pollution in different regions in combination with other influencing factors, such as urban planning, geographical conditions or population density, so as to provide some scientific basis for prevention and control of air pollution. It provides an effective method for exploring the causes of air pollution.

Due to the recurrence of the COVID-19 in the past two years, people have greatly reduced their production and living activities, and affected by which, the air quality is significantly improved, but the economic development is hit hard. Therefore, how to reduce air pollution as much as possible while ensuring economic development is the focus of our attention. The economic development mode of China will not change greatly in a short time, so we are committed to finding a better way to study the mechanism of air pollution, and make our own efforts for improving the atmospheric environment. In the future, whether the air pollution has an impact on extreme weather deserves our attention. And whether the continuous air pollution in Zhengzhou has an impact on the 7.20 severe rainstorm last year, is a subject worthy of study also.

## Figures and Tables

**Figure 1 ijerph-19-08224-f001:**
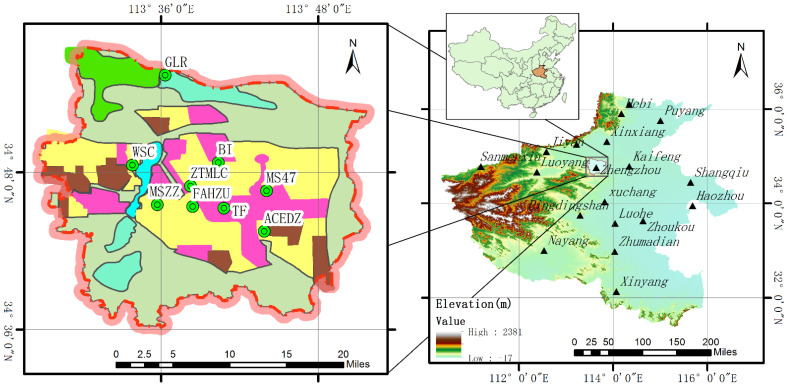
The distribution of automatic monitoring stations in Zhengzhou City and the location of Zhengzhou City in Henan Province.

**Figure 2 ijerph-19-08224-f002:**
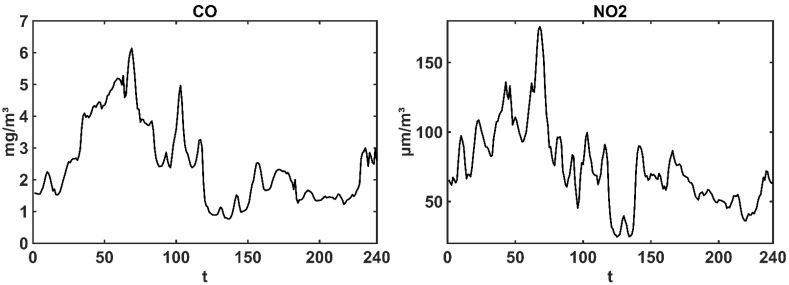
Hourly average NO_2_, CO, O_3_, PM_10_, PM_2.5_ and SO_2_ data of Zhengzhou over the period from 24:00 17 December 2016 to 23:00 26 December 2016 (240 h).

**Figure 3 ijerph-19-08224-f003:**
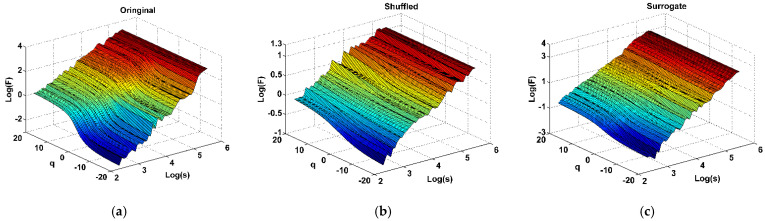
The surface of fluctuation functions $ln(F_{x_{m}^{1},\ldots ,x_{m}^{n}}(q,s))$ versus time scale *ln*(*s*) and moments *q* for (**a**) original pollutants time series, (**b**) shuffled pollutants time series and (**c**) surrogate pollutants time series in Zhengzhou, where *q* ranges from −20 to 20.

**Figure 4 ijerph-19-08224-f004:**
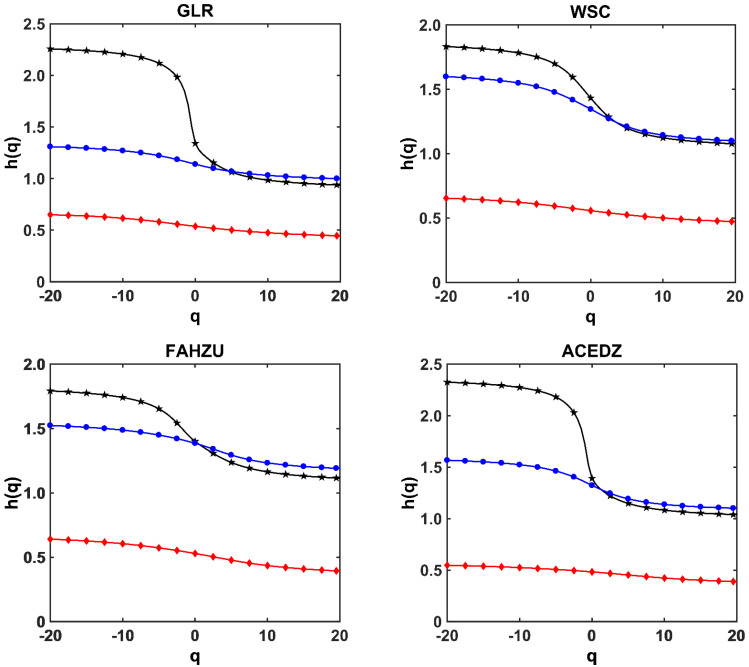
The behaviors of *q*~*h*(*q*) of original series, shuffled series and surrogate series of 6 pollutants at 9 monitoring stations in Zhengzhou using CDFA method.

**Figure 5 ijerph-19-08224-f005:**
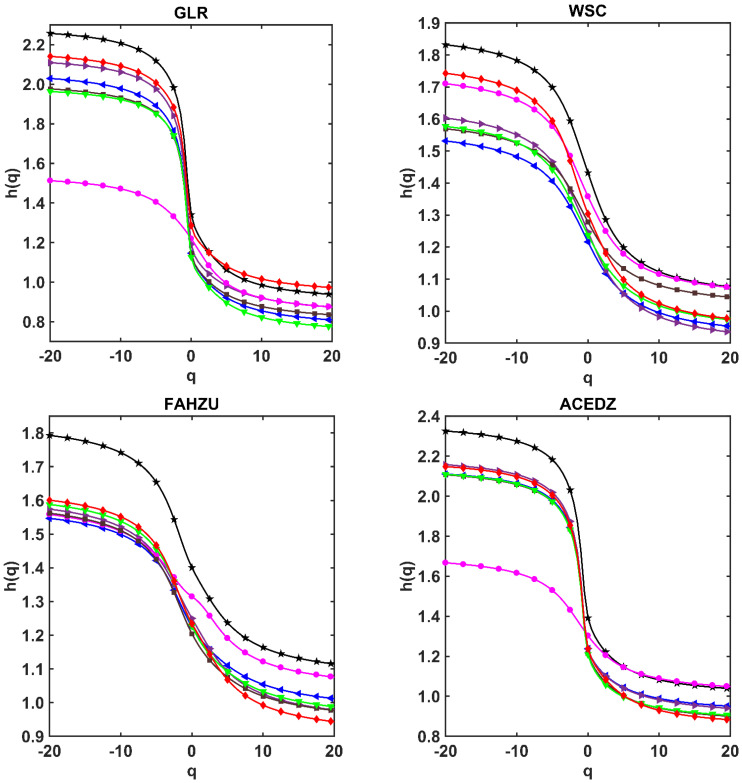
The behaviors of *q*~*h*(*q*) of 6 pollutants in nine monitoring stations in Zhengzhou after CDFA analysis, only the mentioned time series is shuffled and the other time series are kept unchanged.

**Figure 6 ijerph-19-08224-f006:**
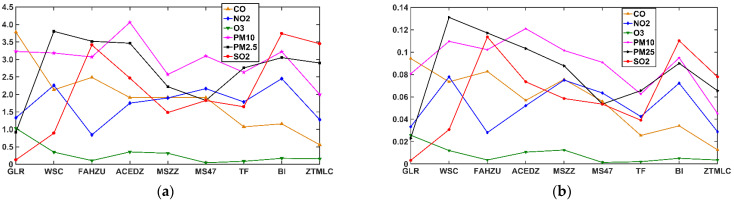
And ${\bar{\chi }}_{shuff}^{2}$ values under CDFA analysis of shuffled series in each station for 0≤q≤20. (**a**) $\chi_{shuff}^{2}$ value in 9 stations, (**b**) ${\bar{\chi }}_{shuff}^{2}$ value in 9 stations.

**Figure 7 ijerph-19-08224-f007:**
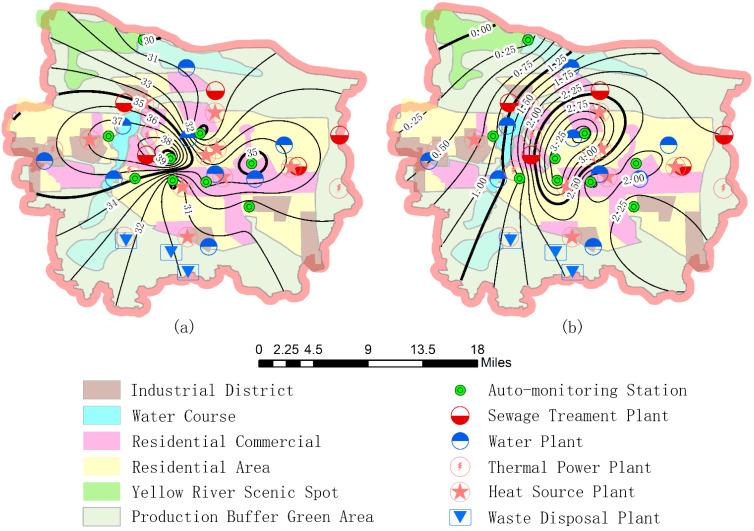
The spatial distribution of (**a**) SO_2_ concentration and (**b**) ${\bar{\chi }}_{shuff}^{2}$ value (0≤q≤20).

**Table 1 ijerph-19-08224-t001:** The average of $\chi_{shuff}^{2}$ and ${\bar{\chi }}_{shuff}^{2}$ values for six pollutants in nine stations.

Ave	CO	NO_2_	O_3_	PM_10_	PM_2.5_	SO_2_
$\chi_{shuff}^{2}$	5.678 × 10^−4^	5.256 × 10^−4^	8.453 × 10^−5^	8.980 × 10^−4^	8.182 × 10^−4^	6.221 × 10^−4^
${\bar{\chi }}_{shuff}^{2}$	1.88	1.751	0.29	3.008	2.719	2.115

## Data Availability

The data used in this study will be provided on request.

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
