# Peer review of "Spatial Characteristics Analysis for Coupling Strength among Air Pollutants during a Severe Haze Period in Zhengzhou, China"

_ijerph, 2022, doi:10.3390/ijerph19148224_

Round 1

Reviewer 1 Report

This manuscript provides a spatial characteristics analysis for coupling strength among air pollutants during a severe haze period in Zhengzhou, China. It merits publication in IJERPH once a number of issues are addressed and the manuscript is extensively revised, including a review of the language and an enhanced discussion of the physical meaning of the methods used and of the results. More specifically:

1. Significant improvements in the use of English language are needed, for example in the abstract and introduction, etc. I recommend review of the language by a native English speaker.

2. Introduction: “as a result of unreasonable human activities such as industrial and agricultural sustained effects,”: this sentence is not understandable. Furthermore, industrial and agricultural activities cannot be characterized as unreasonable per se…

3. Figure 1 should also show the main pollution sources in Zhengzhou, such as industrial sites and main road arteries.

4. It is not clear from which station/s the results of Figure 1 originate. This should be clearly stated in the legend of the figure and/or the text of the manuscript.

5. In section 3.1 some narrative text on the CDFA approach should be added to provide some physical explanation.

6. There are some editorial mistakes in the text. For example, “0” appears in several locations where it should not. On the same time the dates of some references are missing from the text. E.g. line 179 “…proposed by Hedayatifar et al. 0,…”.

7. Lines 203 and 204: should the reference be to figures 2(a) and 2(b) or to figures 3(a) and 3(b)?

8. What is the physical meaning of the analysis provided in sections 4, 4.1 and 4.2 (figures 3, 4 and 5)? What does it really tell us about the concentrations? This should be more clearly presented, for the paper to be more useful for air pollution scientists and policymakers.

9. What are the differences of the data provided in figures 4 and 5, why these two steps were taken? This should be further clarified in the text.

10. Lines 265-267: SO2 is not the only pollutant attributable to central (or other type of) heating. CO, PM10, PM2.5 are also such pollutants. The text here needs to be rephrased accordingly.

11. Lines 283-289: the text is not very clear. It should be expanded to explain the physical meaning of these findings, and what the different values of each pollutant mean. Also, the findings should be discussed in relation to the literature.

12. Line 257: “Zhengzhou is a mainly soot-based pollution city,…”:

(a) Why is that? References or other explanations should be provided!

(b) If it is a soot pollution city, PM10 (and PM2.5) should be a key pollutant, even more than SO2. This should be further explained…

(c) following the above, the selection of SO2 as the key pollutant is not adequately justified in my view. Therefore, the justification should be enhanced.

13. Conclusions, Lines 387-388: “From the perspective of spatial analysis of coupling strength for air pollutants will provide a new theoretical approach for pollution control.”.

Why is that? This statement needs to be explained.

This statement is also repeated in lines 397-400.

Reviewer 2 Report

Dear Authors,

I have the following comments to your manuscript:

Page 1, line 38, 39

in sentence "There are many scholars have used the two methods to study the air pollution time series, such ... "

please rewrite, the meaning is unclear. Is should be like

"Many scholars were used different methods to study the air pollution time series, such ..."

Page 2, line 51, 52

the same as above. Please rewrite.

Page 3, line 121,122

"(http://www.zzepb.gov.cn/)"

move web link into the references section

Page 4, Figure 1

Please explain the location of the monitoring stations in the text. Where are located?

Like next to highways, traffic intersection, parks, habitation etc.

Page 4, Figure 2

Increase graph size, font size, label size, increase quality of the graphs

suggestion, place only two graphs next to each other

Page 6, Figure 3

the same as for Figure 2, above

increase figures size, labels, text

Mark 1,2,3,4,5,6 (six) times series in the graphs by arrows or etc.

Page, 7, Figure 4

increase the font size for legend (star, circle...) above the figure caption.

explain in the body of the text the meaning of the Figure 4 in detail, what does the individual functions (green, blue, red) denotes

Page 8, Figure 5

the same as above for with the previous figures

font size, legend size, figures size

page 9, line 285, 286

in the sentence " It is means that the coupling correlation "

the meaning is unclear please rewrite

like " It means that .... "

Page 12, chapter - Conclusion part

When the city is producing large volume of the emissions, pollution due to industry, vehicles etc. It is very important to make steps to reduce these emissions. Not to increase the volume of emissions; or it is also not smart to ignore the whole situation. This is because in cities live many people and these people hope that also government and scientist perform the maximum effort to reduce these pollutant emission problems.

Therefore I do not agree with your "theory" about "high complexity" and "so-called unique robustness of complex systems".

With such as theory we can do nothing only measure and monitor emissions and current situation and write report(s) that everything is so robust and complex .....?

It is like with spreading of fire around the city. You do not simply ignore the situation with argument that everything is so robust and complex. But we do steps and initiative to minimise the spreading of fire as well as stop the fire!

So please do not try to convince the readers about "complex systems" theory of emissions. Instead do every effort as environmental scientist to minimise these emissions somehow.

At the end of the manuscript include list of all abbreviations mention in the text.

Reviewer 3 Report

Comments on “Spatial characteristics analysis for coupling strength among air pollutants during a severe haze period in Zhengzhou, China”

Air pollution is not only the focus of atmospheric chemists, but also the concern of scholars in various fields. Mathematical and statistical methods, such as the coupling detrended fluctuation analysis (CDFA) method used in this study, are also encouraged to study the relationship between air pollutants. Mathematical or statistical methods have advantages in studying the relationship between pollutants on a long-time scale. In this study, the spatial characteristics analysis achieved by ArcGIS is valuable for local air pollution prevention and control. However, there are several flaws in the present work, as follows:

 1. Autumn and winter are the seasons with frequent haze events, especially the heating period. However, the time span of this study is only 9 days, which is neither representative nor statistically significant. Why not conduct a long-term time series (at least a year) analysis since it is easy to obtain the hourly concentration of air quality data from China National Environmental Monitoring Centre?

 2. Unfavorable meteorological conditions are also one of the main factors leading to haze in autumn and winter. The influence of meteorological factors (wind direction, wind speed, etc.) should but not considered in this study.

 3. From this manuscript, SO2 was identified as the greatest impact in the haze event. SO2 is a primary pollutant, and its corresponding secondary pollutant is sulfate, which is an important component of PM2.5. What is the proportion of sulfate to PM2.5 during heavy pollution events in Zhengzhou? How about nitrate? There is no literature review to make comparison and lacks in-depth discussion and analysis in this part. The conclusion lacks evidence and is unconvincing.

Round 2

Reviewer 1 Report

The last sentence of the abstract "The spatial analysis of coupling strength for air pollutants will provide a new theoretical approach for pollution control. " is not factual and should be changed to reflect the relevant changes done in the conclusions following this reviewer's comments on the original manuscript.

Author Response

The author's response: Thanks for the referee’s good advice. We have modified the last sentence of the abstract as “The spatial analysis of coupling strength for air pollutants will provide an effective approach for pollution control.”

Reviewer 3 Report

I read the revised manuscript and the file of authors’ response to reviewer carefully. I still insist that using mathematical or statistical methods to study the relationship between air pollutants should be based on a large data set. A 9-day data set is the most serious shortcomings of this manuscript.

Another serious problem need to be emphasized again:

In the manuscript, the authors draw a conclusion that SO2 plays a key role in the air pollution of Zhengzhou. SO2 is a primary pollutant, and its corresponding secondary pollutant is sulfate, which is an important component of PM2.5. What is the proportion of sulfate to PM2.5 during heavy pollution events in Zhengzhou? How about nitrate? It was reported that the proportion of nitrate increased significantly during haze events in many urban areas. There is no literature review to make comparison and lacks in-depth discussion and analysis in this part. The conclusion lacks evidence and is unconvincing. Literature review is requested here to support the conclusion that SO2 was identified as the greatest impact in the haze event.

The authors have conducted research on long-time series of air pollution, and have made some achievements. I suggest that the authors integrate the results of a long-time series and resubmit. 
